# Low-Frequency Noise Characteristics of (Al, Ga)As and Ga(As, Bi) Quantum Well Structures for NIR Laser Diodes [note 1]

**DOI:** 10.3390/s23042282

**Published:** 2023-02-17

**Authors:** Simona Armalytė, Justinas Glemža, Vytautas Jonkus, Sandra Pralgauskaitė, Jonas Matukas, Simona Pūkienė, Andrea Zelioli, Evelina Dudutienė, Arnas Naujokaitis, Andrius Bičiūnas, Bronislovas Čechavičius, Renata Butkutė

**Affiliations:** 1Institute of Applied Electrodynamics and Telecommunications, Faculty of Physics, Vilnius University, Saulėtekio Av. 3, LT-10257 Vilnius, Lithuania; 2Department of Optoelectronics, Center for Physical Sciences and Technology, Saulėtekio Av. 3, LT-10257 Vilnius, Lithuania; 3Institute of Photonics and Nanotechnology, Faculty of Physics, Vilnius University, Saulėtekio Av. 3, LT-10257 Vilnius, Lithuania

**Keywords:** electrical noise, fluctuation, optical noise, laser diode, parabolic QW, rectangular QW

## Abstract

Fabry–Perot laser diodes based on (Al, Ga)As and Ga(As, Bi) with single or multiple parabolic or rectangular-shaped quantum wells (QWs) emitting at the 780–1100 nm spectral range were fabricated and investigated for optimization of the laser QW design and composition of QWs. The laser structures were grown using the molecular beam epitaxy (MBE) technique on the *n*-type GaAs(100) substrate. The photolithography process was performed to fabricate edge-emitting laser bars of 5 μm by 500 μm in size. The temperature-dependent power-current measurements showed that the characteristic threshold current of the fabricated LDs was in the 60–120 mA range. Light and current characteristics were almost linear up to (1.2–2.0) *I*_th_. Low-frequency 10 Hz–20 kHz electrical and optical noise characteristics were measured in the temperature range from 70 K to 290 K and showed that the low-frequency optical and electrical noise spectra are comprised of 1/*f* and Lorentzian-type components. The positive cross-correlation between optical and electrical fluctuations was observed.

## 1. Introduction

Laser diodes (LDs) nowadays are widely used in the communication, scientific, and medicine fields and other applications, not only as an effective source for solid-state laser pumping, where signal-to-noise ratio is important [1,2], or as a light source in fiber systems [3], but also in wireless optical communication systems including visual light communication systems, where coherent light is employed as an information carrier [2,3]. The infrared (IR) spectral region is rich in organic and often harmful molecule footprints, such as methane, carbon dioxide, nitrogen, etc. Thus, highly selective sensing systems for environmental pollution monitoring are in demand. Although the large number of publications are focused on the fabrication and investigation of lasers operating in the mid-IR (the characteristic absorption of molecules is more intensive there), commercial detectors convenient for integration into the sensing system are not sufficiently sensitive. Meanwhile, the parameters of the detectors operating in the near IR (NIR) are more attractive: the sensitivity is a hundred times higher in magnitude. Moreover, NIR radiation is friendly to human organisms, and spectroscopic investigation can provide information about skin tissue state, blood analytes, glucose, oxygen, and hemoglobin concentration. Since noninvasive preventive medical, chemical, and biological sensors, where light is used to excite single-molecule quantities and tissues, are used, the stringent requirements for the light source are met [4,5,6]. Despite the wide variety of laser diodes already being commonplace in various applications, scientists are constantly searching for new ways to improve their characteristics in order to achieve better stability of the parameters and longer lifetime, and to expand the field of laser application. Therefore, new material systems and new solutions for the active region design are constantly sought.

GaAs and (Al, Ga)As quantum well (QW) structures are intensively investigated and widely used for the fabrication of near-infrared light sources. Although their technology is well developed, applications of the GaAs- and (Al, Ga)As-based LDs are held back by losses caused by the Auger non-radiative recombination and temperature-sensitivity of the radiation wavelength. In our previous works, the advantages of the introduction of Bi into the GaAs lattice by substituting As for optoelectronic devices were demonstrated [7,8]. It is well known that Bi presence modifies the GaAs bandgap leading to the partial suppression of the non-radiative Auger recombination: with incorporation of Bi atoms into the GaAs lattice, the valence band is mainly modified, while the change in the conducting band is significantly weaker. Moreover, for the larger Bi content in the lattice, the spin-orbit splitting energy increases with a decrease in the energy bandgap, and for the Ga(As, Bi) containing 10% of Bi, the spin-orbit splitting energy overcomes the bandgap value in such a way that it eliminates the Auger losses. On the other hand, the weaker band gap sensitivity to the temperature of the Ga(As, Bi) enables a more stable operation of the Ga(As, Bi)-based LD without external cooling. However, the introduction of Bi into the GaAs system also has several disadvantages: technological problems related to the low-temperature growth process, Bi surface segregation, formation of Bi clusters, and CuPt-type Bi distribution [8,9,10]. To facilitate Bi incorporation and avoid segregation, the structures need to be grown at low temperature and with significantly reduced arsenic fluxes. However, such atypical epitaxy conditions weaken the radiative recombination process and affect the optical properties of the structure. Therefore, attempts to fabricate (Al, Ga)As-based laser diodes with similar characteristics as LDs based on Ga(As, Bi) QWs varying the quantity of Al (from pure GaAs to (Al, Ga)As with up to 10%Al) in the QWs and the profile of the barriers have been carried out [7]. Experimental investigation and modeling of optoelectronic devices with parabolic (and other non-rectangular) quantum structures are quite often shown in the scientific literature [7,11,12,13,14,15,16]. Enhancement of the photoluminescence (increase in the radiating transition probability and efficiency) in structures with parabolic QWs (PQWs) in comparison with the rectangular ones is demonstrated: grading of the barrier layer doping pulls electron and hole wave functions into a narrow region, which reduces their spatial separation in the QW [13,14,15]. This effect is related to the increased carrier trapping efficiency and pronounced localization effect in the GaAsBi PQWs [14,15]. The carrier localization and increase in the trapping efficiency in the GaAsBi QWs is responsible for the observed enhancement in the radiative properties of the parabolically graded barriers’ (PGBs’) structures [7,17]. Therefore, radiating devices with PQWs are more efficient than those with conventional rectangular QWs [13,14]. References also report on a weak function of the carrier density [15] and temperature [7] of the transition energy (hence the wavelength) in the case of the non-square QW structure, while the transition energy monotonically increases (the wavelength decreases) with the temperature in the rectangular QWs. The peak wavelength stability is highly important in many LD applications, especially in communications.

The analysis of the low-frequency noise characteristics of the double-heterostructure QW laser diodes shows that generation and recombination and 1/*f* fluctuations are due to the charge carrier capture centers formed by defects in the active region, and also at the surface and interfaces, which are characteristic for such LDs [8,18,19]. Our previous study, published in ref. [8], concentrated on an investigation of the detailed electrical noise characteristics of standard Fabry–Perot laser diodes containing three rectangular GaAsBi/GaAs quantum wells. The study revealed that defects involving Bi atoms in the GaAs structure are responsible for the formation of carrier trapping centers. Noise characteristics not only clear up processes related to the number of free charge carriers in the active region of the device but also enable detection and analysis of mode-hopping processes in laser diodes [20,21]. Ref. [1] notes that white noise in laser diodes can be attributed to longitudinal or transverse mode partitions. Our previous works also show that during the longitudinal mode hopping, intensive impulse noise with a white spectrum in the limited frequency range is observed both in optical and electrical fluctuations [20,21].

Taking into account our achievements of optical efficiency enhancement in the parabolic design of quantum wells of Ga(As,Bi), our further investigation was implemented by extending the variety of LDs. For this study, two series of laser diodes were fabricated: rectangular and parabolic single or multiple Ga(As,Bi) QWs were used as the cavity in the LDs. The LDs with different designs of QW were investigated in order to reveal the influence of shape. To compare LD parameters, the classical (Al, Ga)As compound was used for QW growth.

Thus, in this paper we present the study of characteristics of optical and electrical low-frequency fluctuations of NIR laser diodes based on, firstly, different material systems, (Al, Ga)As and Ga(As,Bi), and secondly, different shapes of the active area: QWs with standard rectangular and parabolic profiles. The main goal is to understand noise origin and carrier transport mechanisms in these structures and test the quality and mode stability of the fabricated lasers. Characteristic 1/*f* and Lorentzian-type low-frequency noise spectra have shown that electrical and optical fluctuations in the investigated lasers are caused by the charge carrier capture and release in the defect-formed trapping centers. The study revealed that current leakage out of the active region (for low current) affects the electrical noise characteristics, while the leakage influence on the output light characteristics above the threshold is insignificant.

## 2. Fabrication of Laser Diodes

The investigated laser diodes were grown using two SVT-A and Veeco GENxplor R&D molecular beam epitaxy (MBE) reactors equipped with standard cells for metallic Al, Ga, Bi, and unique arsenic source design generating pure arsenic dimers flux. The *n*-type GaAs substrate oriented in a (001) crystalline plane was used for the deposition of the laser structures. Prior to the growth, the native oxide from the substrate was outgassed at 700 °C substrate temperature for 30 min supplying maximum arsenic flux. The silicon-doped GaAs buffer layer (2 × 10^18^ cm^−3^) of 50–100 nm thickness was deposited to provide a flat surface for epitaxial growth. Device structures were based on single or multiple QW structures of (Al, Ga)As and Ga(As, Bi) compounds (see Figure 1). The number of QWs varied from one to two or three wells, respectively, for classical (Al, Ga)As and highly disordered Ga(As, Bi) semiconducting compounds (Table 1). To reach the emission wavelength in the range of 780–1100 nm, both the width and composition of the QW were changed. The (Al, Ga)As QW width varied from 1.5 nm to 5.0 nm. While the thickness of the Ga(As, Bi) QW varied from 7.0 nm to 12.0 nm, respectively, for the Ga(As, Bi) with 6% and 4% of Bi. LDs with (Al, Ga)As QWs were grown under standard conditions: 650 °C temperature and arsenic overpressure. To introduce more Bi into the Ga(As, Bi) QW and retain high optical laser diode performances, the two-temperature regime was chosen, i.e., decrease in temperature down to 350–425 °C for the growth of Ga(As, Bi) quantum wells allowed to control the Bi incorporation, and the waveguide layers growth at the standard high temperature assured a high crystalline quality. It was supposed that the high-temperature upper (Al, Ga)As *p*-type clad ding layer growth serves as an annealing of the bismide QWs, which could reduce the disorder of the alloy and improve bismuth distribution in the QW. The more detailed technological parameters of the epitaxy of the laser diodes containing Ga(As, Bi) are published in [22].

To determine the effect of the QW geometry on the LD operation and noise characteristics, structures with two quantum well profiles, conventional rectangular (RQW) or parabolic profile (PQW), were grown and investigated. From the technological point of view, the PQWs can be realized by gradually changing the Al content by the quadratic law in the defined layer or by inserting (Al, Ga)As barrier layers with constant Al content yet of different thicknesses following the parabolic law in the GaAs. In [7], the modeling and growth process of the grading profile of PQW is described. The active region was surrounded by 200 nm (Al, Ga)As spacer layers with 30% of Al. The 1.5 μm thick (Al, Ga)As:Si and (Al, Ga)As:Be layers (with a doping level of 2 × 10^17^ cm^−3^), both with 55% of Al, served as waveguides. To improve the metal–semiconductor contact the 50 nm thick highly *p*-doped (3 × 10^18^ cm^−3^) GaAs was deposited on top of the (Al, Ga)As:Be layer. The schematic picture of LD and various designs of the active area containing PQW and RQW of (Al, Ga)As QW and Ga(As, Bi) QW are presented in Figure 1. The main QW parameters of the growth and investigated structures in this work are summarized in Table 1.

The ~500 μm length and 5 μm wide laser bars were processed by UV lithography. The Ni–Au metal contact was deposited using the e-beam deposition technique on the top of the epitaxial structure through an opening in the photoresist layer. The Au–Ge metallic layer deposited on the bottom of the structure was used as an *n*-type ohmic contact. Figure 2 shows a SEM picture of the view of the LD bar after UV lithography and metallization (a) and a zoom-in image (b) with red dashed lines indicating the details of the textured structure as a visual guide.

The evaluated lasing threshold current for different laser diodes was 40–60 for (Al, Ga)As PQW and RQW mA (in cw mode), 80 mA for Ga(As, Bi) PQW, and 120 mA for Ga(As, Bi) RQW (the LDs containing Ga(As, Bi) QW were measured using pulsed bias of 100 ns duration and 1 kHz repetition rate electrical pulses).

Table 1 presents the main technological and electroluminescence measurement data of the investigated laser diodes. The activation energies of the charge carrier trapping centers for different laser diode structures were established from the temperature characteristics of the noise spectra (see Section 4).

## 3. Noise Measurement Technique

Electrical and optical fluctuations of laser diodes were measured at low frequencies of 10 Hz–0.20 MHz. Electrical noise is the laser diode voltage fluctuation, while optical noise represents the fluctuation of the laser output light power (LD output light was detected by Ge photodiode: optical noise corresponds to fluctuation of the photodiode voltage due to LD output power fluctuation). Moreover, the cross-correlation factor between these two noise signals gives valuable information on the charge carrier transport in the laser diode structure [19,20,23] and was measured in the 10 Hz–20 kHz frequency range as well as in 1 octave frequency bands [19].

Voltage noise signals were measured under constant current mode in a forward bias at room temperature and in the temperature range from 71 K to 290 K. Liquid nitrogen was used to cool down the sample chamber and a thermoelectric cooler was used for stabilization of the sample temperature (to suppress the self-heating effect). The measured noise signals were amplified by the low-noise amplifier, passed through a filter system, converted into a digital signal (by National Instruments PCI-6115), and analyzed by a personal computer [19]. Specially designed self-made low-noise amplifiers based on parallel-connected 2SK117 n channel J-FETs, which enable reduction in the noise equivalent resistance to 13 Ω at 1 kHz, were used for the first stage of amplification (bandwidth is up to 1 MHz, amplification is 20 times). Decoupling capacitors are used in amplifiers to block the DC component of the signal. For the next stage, the SR560 low-noise voltage amplifiers, which have a built-in filter system of up to 1 MHz and their gain varies from 1 to 50,000, were used. Amplified noise signals go to the NI BNC-2110 shielded terminal block which is connected to the PCI-6115 analog–digital converter by a shielded cable. Noise measurements were carried out in a specially shielded laboratory room (the Faraday cage), which made it possible to avoid parasitic interference from ambient electromagnetic radiation. The measuring setup and the device under test were biased from autonomous batteries in order to avoid additional disturbances from a power network.

The spectral density of the voltage fluctuation is estimated by using Cooley–Tukey fast Fourier transform (FFT) and calculated spectral densities were averaged by using over 300 realizations in a narrow frequency band Δ*f* = 0.1*f* (here *f* is the central frequency of Δ*f*). The Palenskis–Shoblitzkas autocorrelation weight window [24], which gives minimum distortion for the low-frequency noise spectrum, was used to smooth the noise spectrum. The absolute value of the measured voltage noise spectral density was calculated by comparison with the thermal noise of the reference resistance, $R_{\mathrm{ref}}$:(1)$$S_{U}=\frac{\bar{\Delta u_{\mathrm{el},\mathrm{opt}}^{2}}-\bar{\Delta u_{s}^{2}}}{\bar{\Delta u_{\mathrm{ref}}^{2}}-\bar{\Delta u_{s}^{2}}}4kTR_{\mathrm{ref}};$$
where $\bar{\Delta u_{\mathrm{el},\mathrm{opt}}^{2}}$, $\bar{\Delta u_{s}^{2}}$, and $\bar{\Delta u_{\mathrm{ref}}^{2}}$ are, respectively, the variance of the laser diode noise voltage (if electrical noise is measured) or the variance of the noise voltage of the photodetector load resistance (if optical noise is measured) plus the noise of the measuring system, the variance of the voltage fluctuation of the measuring system, and the variance of the reference resistor thermal noise plus the noise of the measuring system in the narrow frequency band Δ*f*; *k* is the Boltzmann’s constant and *T* is the absolute temperature of the reference resistor.

The measuring setup is comprises two identical channels: one channel for electrical noise and another for optical noise measurement. Simultaneous measurement of both optical and electrical noise signals enables calculation of the cross-correlation factor with an accuracy of 1% [21]:(2)$$k=\frac{\bar{\Delta u_{\mathrm{el}}\cdot \;\Delta u_{\mathrm{opt}}\;}}{\sqrt{\bar{\Delta u_{\mathrm{el}}^{2}}\cdot \bar{\Delta u_{\mathrm{opt}}^{2}}}};$$
where $\Delta u_{\mathrm{el}}$ and $\Delta u_{\mathrm{opt}}$ are the voltage of electrical and optical, respectively, noise signals. A positive cross-correlation factor indicates that optical and electrical noise signals change simultaneously in phase, while negative correlation indicates a change in opposite phases.

## 4. Results

### 4.1. Typical Operating Characteristics of the Investigated Laser Diodes

Operation characteristics: current–voltage (IV) and light output power vs. current (LI) of the investigated laser diodes are presented in Figure 3. The IV characteristic of the lasers with both types of quantum wells—rectangular and parabolic—are typical for *pn* junction diodes, as well as LI characteristic. The threshold current (*I*_th_) of the fabricated lasers varies from 40 mA to 120 mA; a more noticeable saturation of LI characteristic is observed for particular samples starting from approximately (1.2–2.0) *I*_th_.

### 4.2. Low-Frequency Noise Spectra

1/*f* and Lorentzian-type spectra are observed in low-frequency noise characteristics of the investigated lasers (Figure 4 and Figure 5). Electrical noise spectra have quite clear Lorentzian-type bumps over the 1/*f* spectrum at the particular bias current and temperature values, while optical noise spectra are almost 1/*f* type over the measured frequency range (at the injection current above the threshold, i.e., when the output light power fluctuations surpass the measuring system noise).

### 4.3. Noise Spectral Density vs. Current Characteristics

For most of the investigated AlGaAs, the GaAs-based lasers’ electrical noise spectral density at small currents was in the range of one order of magnitude (characteristics for some of these samples are presented in Figure 6; curves for other samples are similar, and partially overlap with the presented ones). However, particular samples demonstrated quite large fluctuations, e.g., sample No. 7, in which the electrical noise spectral density was larger by about 2–3 orders of magnitude both at small currents and at the lasing operation. This sample also has a large threshold current (102 mA (compared with ~70 mA for the same structure sample No. 6, for which characteristics are presented in Figure 3, Figure 4 and Figure 5)) and large positive cross-correlation between optical and electrical fluctuations (Figure 7). The electrical noise spectral density of the Ga(As,Bi)-based lasers (e.g., sample No. 2 in Figure 6) at small currents is also 1–2 orders of magnitude larger comparing with the AlGaAs and GaAs ones (Figure 6a), which indicates the larger defectivity of the Ga(As,Bi)-based devices.

Optical noise intensity of the investigated lasers has a sharp peak at the threshold current as a reaction to the change in the operation mode from the light-emitting diode mode to the lasing one (Figure 7). Optical noise starts to increase at lower currents before the threshold and this increase is larger for the lasers with parabolic QWs than for the rectangular QWs structures. The investigated lasers have smaller or larger positive cross-correlation between optical and electrical fluctuations above the threshold (Figure 7b).

### 4.4. Mode Hopping

The longitudinal mode-hopping effect was observed in the sample No. 4 operation around 90 mA at room temperature (Figure 8 and Figure 9). Mode hopping is an abrupt change in the radiating mode at the particular operation conditions (injection current and temperature) when random hopping between two longitudinal modes (or mode sets) occurs (Figure 9b). In the low-frequency noise characteristics, the mode hopping manifests by strong Lorentzian-type spectra primarily in the optical fluctuations (Figure 8a) but can also be observed in the electrical noise spectra. Furthermore, the strong positive or negative cross-correlation between the optical and electrical fluctuations is characteristic of the mode-hopping effect (see curve at 90 mA in Figure 9a: here, normally the moderate positive value of the cross-correlation is decreased due to the intensive negatively correlated Lorentzian-type noise component). It is shown that some hysteresis of the mode-hopping effect takes place concerning the injection current and temperature [20,21].

## 5. Discussion

Lorentzian-type spectra are characteristic when there is a single charge carrier trapping center that is more active compared with others. While when there are many trapping centers of similar activity and with a wide distribution of characteristic times, 1/*f* type spectra are observed. The obtained noise characteristics indicate that low-frequency fluctuations in the investigated lasers are caused by the charge carrier capture and release processes through the defect-formed trapping centers. These processes lead to fluctuation of the charge carrier density and, therefore, fluctuation of the resistance.

From the cross-correlation factor dependency on the frequency (Figure 5), it is clearly seen that only 1/*f* type noise components in electrical and optical fluctuations are correlated at low frequencies. The cross-correlation factor increases with the injection current due to the rise in the optical noise intensity and drops at frequencies above 1 kHz (where the Lorentzian-type bumps prevail over the 1/*f* spectrum of electrical fluctuations). The presence of the Lorentzian-type bumps only in the electrical noise characteristics suggests that these generation and recombination fluctuations originated from the carrier trapping in the centers that are not in the active region, however, they cause an injection current leakage around it. This influences the laser threshold current and electrical noise characteristics but is not reflected in the manner of the output light characteristics. The high-threshold current of Ga(As, Bi) QW-based lasers was discussed in the review of I. P. Marko [25]. It was demonstrated that threshold current depends on the Bi concentration in the quantum well; with an increase in the Bi content, the threshold current increases linearly. Authors concluded that for the delivery of better performances of the laser characteristics, optimization of the device design with optimized QW thickness, alloy composition, strain and band offsets, as well as QW number and design, is necessary.

The previously mentioned carrier trapping centers are active at the particular operation conditions: temperature and current. The activity of the center is the largest when the Fermi level coincides with the trapping center energy. As mentioned, the Lorentzian-type spectrum is characteristic for noise when a particular center is active. The frequency, $f_{0}$, at which the “white” level of the spectral density in the Lorentzian spectrum drops twice corresponds to the effective relaxation time of the charge carriers ($\tau =\frac{1}{2\pi f_{0}}$), which is proportional to the activation energy of the trapping center [26,27]. As the activity of the carrier trapping center depends on temperature, the relaxation time dependency on temperature enables the evaluation of the activation energy of this center. Figure 10a shows an example of such an evaluation for one of the observed generation and recombination processes. For a better setting of the $f_{0}$ frequency, the noise spectral density was multiplied by the frequency that causes the curve maximum at $f_{0}$. The relaxation time of the observed trapping centers varies between 1 ms and 10 ms. The dependency of the relaxation time on the temperature of this center is presented in Figure 10b. These dependencies enable the evaluation of the active energy of the centers; the activation energy of these centers varies from 0.03 eV to 0.18 eV (Table 1). Similar values of the activation energy, 33 meV, 80 meV, 109 meV, 132 meV, and 172 meV, were established for the AlGaAs/GaAs heterostructures in [28,29]. Carrier trapping centers with such activation energies are caused by the defects (antisites AsGa and deep donor DX centers) in (Al,Ga)As layers with different Al content, and also in (Al,Ga)As doped with silicon. The larger activation energy corresponds to the larger Al content in (Al,Ga)As spacer layers and silicon-doped waveguides. The activation energies below 100 meV can be explained by the thermal activation of the barrier defects. Since Ga(As, Bi) is a highly disordered alloy, bismuth atoms’ incorporation is complicated. In an ideal case, bismuth must replace arsenic atoms, but at low growth temperatures, arsenic antisite defects are very likely. The second reason for the larger intensity of electrical fluctuations of Ga(As,Bi) QW-based LDs could be the CuPt-type Bi atom distribution. Bi-related defect pairs (AsGa + BiGa) may also be responsible for the carrier generation and recombination centers with 180 meV activation energy as indicated in [30].

The observed electrical noise intensity variation with forward current is typical for laser diodes [8,18,23]; it is large at small currents and decreases up to the particular current value in inverse to the current and then increases with further current increases (see Figure 6). Large electrical noise at small currents is caused by the defects that form in current flow channels and, while current density is small, the current does not flow through the whole cross-section of the structure but through the separate defect-formed channels.

The positive cross-correlation factor shows that there are defects in the active region, which modulate current flow through the active region. For example, larger positive cross-correlation is characteristic of PQW sample No. 7, which also has a larger electrical noise intensity and larger threshold current (Figure 7). It is observed that the optical noise spectral density and cross-correlation factor of lasers with PQWs increases more strongly with the injection of current before the threshold than the characteristics of RQW-based lasers. This indicates more intensive spontaneous radiation in PQW structures, but this does not affect their lasing operation characteristics.

The mode-hopping effect was observed during the operation of some investigated lasers at particular operating conditions. The mode-hopping process is controlled by defects that randomly change the number of the free charge carriers in the active region and cause intensive optical and electrical fluctuations with a characteristic Lorentzian-type spectrum [20,21]. For example, a clear Lorentzian-type spectrum is observed in optical fluctuations at 90 mA in characteristics of sample No. 4, while at the “not mode hopping” region, spectra are 1/*f* type (Figure 8 and Figure 9). Nevertheless, the electrical noise spectra even at the mode hopping are 1/*f*, but the electrical noise spectral density for this sample is about two orders of magnitude larger compared with the other investigated lasers (Figure 4). Therefore, the effect of defects in the active region of this sample causes large 1/*f* fluctuations in the charge carrier number, which determines which longitudinal mode is radiated (Figure 9b shows how the laser radiation spectrum changes with a small current shift). Electrical and optical fluctuations due to mode hopping are negatively correlated (Figure 9). The characteristic positive cross-correlation at 90 mA is decreased by the negatively correlated mode-hopping component in the frequency range between 1 kHz and 10 kHz, where the Lorentzian-type bump is observed in the optical spectrum.

## 6. Conclusions

The operation and low-frequency noise characteristics of near-infrared laser diodes based on (Al,Ga)As and Ga(As,Bi) rectangular and parabolic quantum wells were investigated. The observed low-frequency optical and electrical fluctuations have 1/*f* and Lorentzian-type spectra and are caused by the charge carrier capture and release processes in the defects formed in the carrier trapping centers. At small currents, current leakage out of the active region is reflected in the electrical noise characteristics but does not influence the output light characteristics above the threshold. The evaluated activation energy of these centers is in the range of 0.03 eV to 0.18 eV. The thermally activated centers with 30 meV and 180 meV activation energies were characteristic for both materials: (Al,Ga)As and Ga(As,Bi) LD structures. The determined activation energies were attributed to the antisite As_Ga_ defects and deep donor DX centers in the (Al,Ga)As spacer as well to the silicon-doped (Al,Ga)As waveguide layers.

Measurements of the low-frequency noise demonstrated that optical noise intensity is comparable for both types of LDs, rectangular and parabolic quantum wells, while for Ga(As,Bi) QW-based LDs, a larger intensity of electrical fluctuations was found, which indicates larger defectivity of these structures. The observed positive cross-correlation of the moderate intensity 1/*f* type optical and electrical fluctuations above the threshold shows that there are also defects in the active region of the investigated lasers; these defects modulate the current flow through the active region and directly affect the emitted light power.

Noise analysis shows that the quality of the investigated laser diodes primarily depends on defects in the structure and noise characteristics do not depend on the quantum well profile—rectangular or parabolic—and Al content in the well. However, the addition of Bi causes larger defectivity of the structure, and, therefore, lower efficiency of the laser.

## Figures and Tables

**Figure 1 sensors-23-02282-f001:**
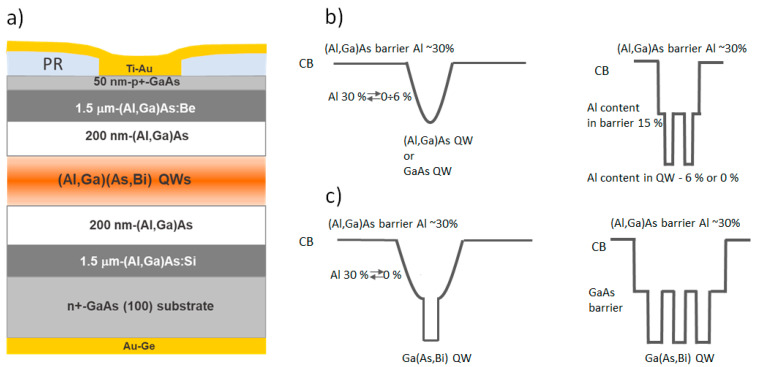
The schematic picture of the laser diode structure (**a**) and various designs of the active area containing parabolic and rectangular single or multiple QWs of (Al, Ga)As and Ga(As, Bi) (**b**,**c**).

**Figure 2 sensors-23-02282-f002:**
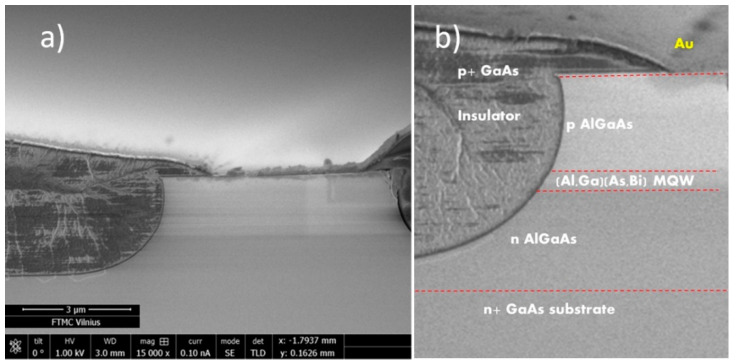
SEM picture of the view of the as-cleaved 5 μm width LD bar after UV lithography and metallization (**a**) and a zoom-in image (**b**) with red dashed lines indicating the details of the textured structure as a visual guide.

**Figure 3 sensors-23-02282-f003:**
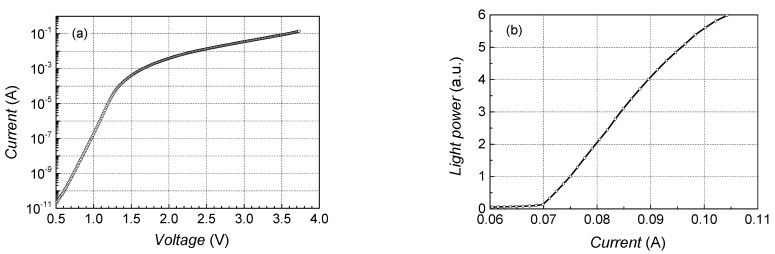
Typical current–voltage (**a**) and light vs. current (**b**) characteristics of the investigated lasers at room temperature (sample No. 6: AlGaAs-based LD with single parabolic QW).

**Figure 4 sensors-23-02282-f004:**
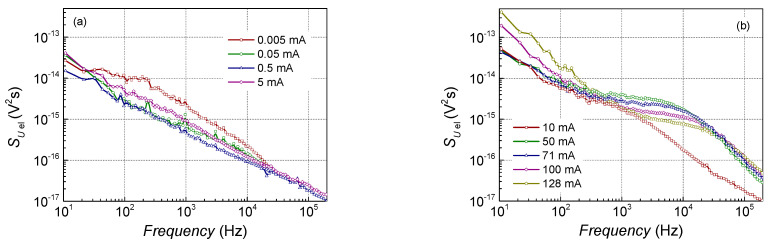
Typical electrical noise spectra of the investigated laser diodes at room temperature at small currents below the threshold (**a**) and at larger current region around and above the threshold (**b**) (sample No. 6: AlGaAs-based LD with single parabolic QW).

**Figure 5 sensors-23-02282-f005:**
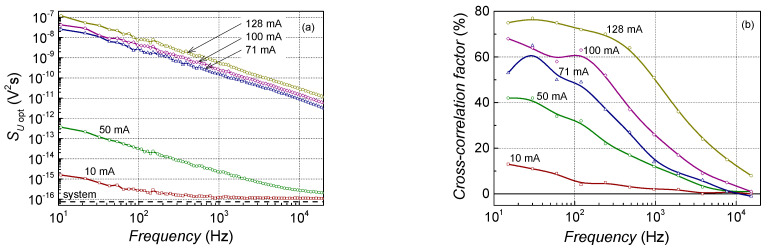
Typical optical noise spectra of the investigated laser diodes at room temperature (**a**) and cross-correlation factor dependency on the frequency at room temperature (**b**) (sample No. 6: AlGaAs-based LD with single parabolic QW; the “system” corresponds to the noise level of the measuring system, which for optical noise measurement is determined by the value of the thermal noise of the photodetector load resistance).

**Figure 6 sensors-23-02282-f006:**
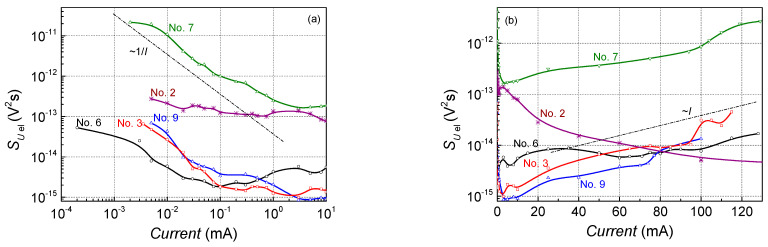
Electrical noise spectral density dependency on forward current at room temperature for laser diodes with parabolic and rectangular QWs at 108 Hz: in the small current region (**a**); at large currents (**b**).

**Figure 7 sensors-23-02282-f007:**
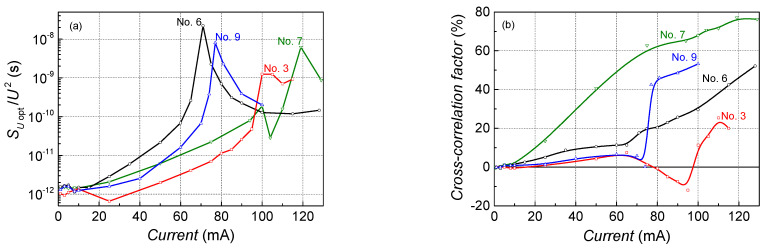
Normalized optical noise spectral density at 108 Hz (**a**) and cross-correlation between optical and electrical fluctuations in the 10 Hz–20 kHz frequency range (**b**) dependencies on forward current at room temperature for laser diodes with parabolic and rectangular QWs.

**Figure 8 sensors-23-02282-f008:**
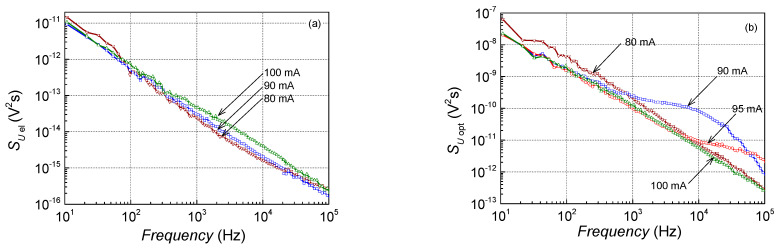
Electrical (**a**) and optical (**b**) noise spectra during the mode-hopping effect of sample No. 4 at room temperature.

**Figure 9 sensors-23-02282-f009:**
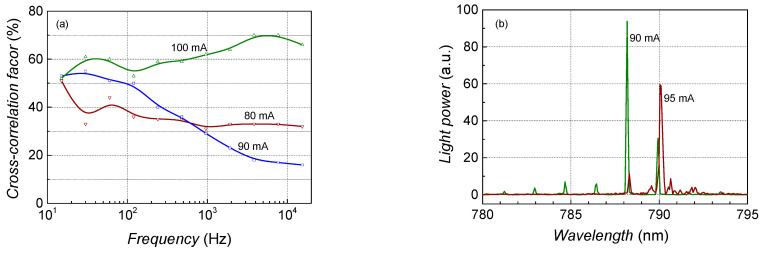
Cross-correlation factor dependency on frequency (**a**) and radiation spectra (**b**) of sample No. 4 during the mode-hopping effect (room temperature).

**Figure 10 sensors-23-02282-f010:**
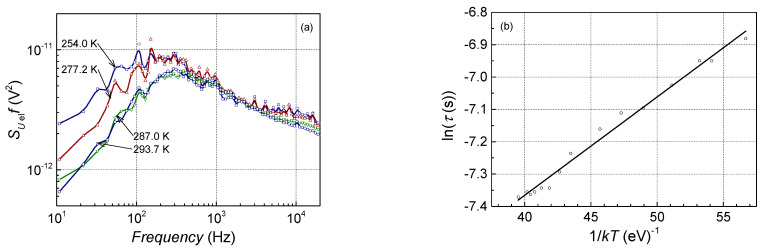
Lorentzian-type electrical noise spectra multiplied by frequency at 0.01 mA forward current (**a**) and characteristic time dependency on temperature (**b**) for the generation and recombination processes observed from 205 K to 294 K in sample No. 9.

**Table 1 sensors-23-02282-t001:** The main parameters of the investigated laser diodes: QW material—quantum well composition; QW shape—QW profile; *N*_QW_—number of QWs; *d*_QW_, nm—QW thickness; *I*_th_, mA—threshold current; *λ*, nm (at RT)—lasing wavelength at room temperature; *E*_a_, meV—activation energy of the trapping centers.

Sample No.	QW Material	QW Shape	*N* _QW_	*d*_QW,_ nm	*I*_th_, mA	*λ*, nm (at RT)	*E*_a_, meV
1	Ga(As,Bi)	Rectangular	3	8	~120	1060	---
2	Ga(As,Bi)	Parabolic	1	10	~80	1030	30; 80; 100; 140; 180
3 4 5	(Al,Ga)As (Al,Ga)As (Al,Ga)As	Rectangular Rectangular Rectangular	2 2 1	1.5 2 5	40–60 40–60 40–60	780 795 820	80, 100, 140, 180 80, 100, 140, 180 80, 100, 140, 180
6	(Al,Ga)As	Parabolic	1	5	40–60	827	---
7	(Al,Ga)As	Parabolic	1	5	40–60	827	---
8 9	GaAs GaAs	Rectangular Rectangular	2 2	1.5 2	40–60 40–60	805 780	30 30

## Data Availability

Not applicable.

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
