# Peer review of "Low-Frequency Noise Characteristics of (Al, Ga)As and Ga(As, Bi) Quantum Well Structures for NIR Laser Diodes"

_sensors, 2023, doi:10.3390/s23042282_

Round 1

Reviewer 1 Report

In this paper, the authors present the investigation on the low frequency fluctuations of NIR laser diodes based on two material systems and different shape of QWs. It is helpful for readers to understand the characteristics of the NIR laser diodes.

Here are some questions that the authors are suggested to improve.

1, the low noise frequency of the NIR laser diodes are old topics, the authors themselves have published similar topics, please enhance the novelty and difference in the part of introduction.

2, please compare the difference of the low-frequency nosie characteristics of the two material systems and explain the reason.

Author Response

Response in the attached file.

Reviewer 2 Report

see the attachment, please.

Author Response

Response in the attached file.

Reviewer 3 Report

The reviewed paper is devoted to quite actual topic development of methods for noise-based characterization of laser diode structures. Interesting approach is related with the study of correlations between electrical and optical fluctuations. However, some aspects of the paper require clarifications and amendments:

1. Experimental equipment had to be described more detailed. Do you use commercial of self-made low-noise amplifier? What characteristics (frequency band, gain coefficient, power spectrum density) does the amplifier have? Which electrical connection is used? The same concerns the filter system. PCI-6115 has the resolution of 12 bit. Is this enough to study electrical noise? Do you use battery or power line supply?

2. Please, describe detrending procedures.

3. Which parameters of spectral density estimation do you use (number of samples, type of window, averaging method)?

4. If understand right, you use equation (1) to estimate spectral density. The equation comprises variances and “white noise” component 4kTR. Please explain how this equation explains 1/f and Lorentzian frequency dependencies of experimental data. Whether the condition mean(u2ref)> mean(u2s) is always met?

5. Calculation of eq. (2) requires simultaneous measurement, how it is implemented?

6. Sample No.2 has different behavior in comparison with other samples (Fig. 6). However, no explanation is provided. There is no information about samples No. 1 and 8.

7. Fig. 5, a provides information about system noise PSD. What system do you mean?

8. Line 251 describes “reaction to the abrupt change of the operation mode”. Please, explain the specific meaning of “abrupt change” in this context.

9. Please, provide explanation and proof for the statement “Maximum dependency of the spectral density multiplied by the frequency corresponds to the characteristic relaxation time of the active center”.

Author Response

Response in the attached file.

Round 2

Reviewer 1 Report

I think the revised manuscript can be accepted for publication.

Reviewer 3 Report

I am think that the introduced corrections makes this article more clear and accurate. I expect that in present form the paper will be intersting to readers.